# Alginate Coating Charged by Hydroxyapatite Complexes with Lactoferrin and Quercetin Enhances the Pork Meat Shelf Life

**DOI:** 10.3390/foods12030553

**Published:** 2023-01-26

**Authors:** Angela Michela Immacolata Montone, Francesca Malvano, Roberta Taiano, Rosanna Capparelli, Federico Capuano, Donatella Albanese

**Affiliations:** 1Department of Industrial Engineering, University of Salerno, 84084 Fisciano, SA, Italy; 2Department of Food Inspection, Istituto Zooprofilattico Sperimentale del Mezzogiorno, 80055 Portici, NA, Italy; 3Department of Agriculture, University of Naples “Federico II”, 80055 Portici, NA, Italy

**Keywords:** active edible coating, hydroxyapatite, quercetin, lactoferrin, cold storage

## Abstract

In this work, the effect of an alginate-based coating loaded with hydroxyapatite/lactoferrin/quercetin (HA/LACTO-QUE) complexes during the storage of pork meat was evaluated. FT-IR spectra of HA/LACTO-QUE complexes confirmed the adsorption of QUE and LACTO into HA crystals showing the characteristic peaks of both active compounds. The kinetic releases of QUE and LACTO from coatings in an aqueous medium pointed out a faster release of LACTO than QUE. The activated alginate-based coating showed a high capability to slow down the growth of total viable bacterial count, psychotropic bacteria count, *Pseudomonas* spp. and *Enterobacteriaceae* during 15 days at 4 °C, as well as the production of the total volatile basic nitrogen. Positive effects were found for maintaining the hardness and water-holding capacity of pork meat samples coated with the activated edible coatings. Sensory evaluation results demonstrated that the active alginate-based coating was effective to preserve the colour and odour of fresh pork meat with overall acceptability up to the end of storage time.

## 1. Introduction

Meat and meat products are important sources of food nutrients such as proteins and B-complex vitamins [1]. However, their composition makes them highly perishable products with a short shelf life. The main phenomenon related to the spoilage of meat is the microbial growth that occurs during storage, causing off-odours and flavours that make the product unsuitable for human consumption [2].

Recently, several approaches have been proposed to preserve the safety and quality of meat and meat products, such as edible films and coatings from biodegradable biopolymers [3].

Among several polysaccharide-based biopolymers commonly used for edible films and coating for food preservation, chitosan and sodium alginate are the most exploited for meat and meat products [4,5]. However, unlike chitosan, which has shown to be an effective natural antimicrobial agent against Gram-positive and Gram-negative bacteria [6], alginate-based layers act only on the water and gas exchanges [5] with no effect on *Pseudomonas* spp. and *Enterobacteriaceae* growth during the cold storage of chicken fillets [7]. Similar results were reported by other authors [8,9], highlighting the same total viable count increment rate during the storage of uncoated and alginate-based coated fresh chicken breast meat.

However, active edible films and coatings loaded with antimicrobial and antioxidant compounds seem to have great potential for preserving the quality and prolonging the shelf life of meat products [10]. Recently, different studies evaluated the ability of active edible coatings enriched with essential oils [11,12,13], phenolic compounds [14], organic acids [15] and natural extracts [16,17] to preserve the quality of fresh meat and meat products during storage. Positive effects of coatings enriched with essential oils [11,12,13], phenolic compounds [14], organic acids [15] and natural extracts [16,17] were obtained in extending the shelf life of meat, inhibiting microbial growth as well as lipid oxidation and weight loss. As regards the application of active alginate-based coatings in fresh meat and different types of meat products, we report in Table 1 some of the published reports in the last 5 years.

Among active compounds, the flavonoids’ quercetin and quercetin glycosides with antioxidant [23] and antimicrobial activity [24] are present in various fruit and vegetables besides being used in numerous consumer applications such as dietary supplements. Lactoferrin is a glycoprotein with well-known antimicrobial activity [25,26]. The use of lactoferrin as a nutritional supplement is GRAS by the US Food and Drug Administration and a novel food ingredient after the positive scientific opinion provided in 2012 by European Food Safety Authority.

Due to the high sensitivity of these compounds to pH, temperature and light, a carrier for their prompt release is necessary to overcome the problem of their use as active compounds in food packaging.

Moreover, the efficacy of an antimicrobial packaging system also depends on the kinetic release of the compound from the coating to the food surface. The latter depends on both the solubility of the compounds in an aqueous medium and the type and strength of the polymer network used to produce the coating. Moreover, to protect the bioactive compounds from degradation that could occur in the edible coating during the storage period, carrier systems such as polymeric nanoparticles [27,28,29], nanoemulsion [30,31,32] and nanocomposites [33] have been proposed in the literature of alginate-based edible film and coatings. Within the delivery systems of active compounds, hydroxyapatite (HA), due to its chemical physical properties including biocompatibility, stability, and degradability, could be an interesting candidate for carriers in food packaging applications. It represents the major component of cartilaginous tissues, such as bone and tooth; due to the biomimetic crystal structure and properties, HA crystals are widely employed in different medical practices [34]. Furthermore, the use of HA in food is allowed in Europe by Regulation (EC) No 1333/2008 on food additives classifying the HA by the code E341.

To the best of our knowledge, no previous studies exist in the literature on the application of hydroxyapatite as a component of edible coatings for food shelf-life extension, except those published by our research group [7,35,36]. Our research group recently developed alginate-based coatings loaded with hydroxyapatite/quercetin complexes for the shelf-life extension of fresh chicken fillets [7]. To increase the antimicrobial activity versus *Pseudomonas* spp. we also evaluated the synergistic effect of lactoferrin and quercetin loaded in hydroxyapatite crystals at different active compound concentrations [25].

Based on these considerations, this work aimed to evaluate the effectiveness of HA lactoferrin and quercetin complexes loaded in an alginate-based coating during the cold storage of fresh pork fillets. For this purpose, the effects of alginate coating charged with free lactoferrin and quercetin and HA lactoferrin and quercetin complexes were carried out by the evaluation of microbiological, physical and sensory properties of pork fillets stored at 4 °C for 15 days. Moreover, the physical characterisation of Hydroxyapatite complexes was performed besides studying the kinetic release of lactoferrin and quercetin from activated alginate-based coatings.

## 2. Materials and Methods

### 2.1. Materials

Fresh pork meat was purchased at a food retailer meat counter (Salerno Italy) and cold (4 ± 1 °C) was transported to the laboratories of the University of Salerno.

Sodium alginate, calcium chloride and glycerol were all obtained from Sigma-Aldrich (Milano, Italy). Quercetin glycoside compounds (QUE, 98.6% food grade) were purchased from Oxford^®^ Vitality Company (Bicester, UK) and Lactoferrin (LACTO, 95% food grade) from Fargon (Newcastle, UK). Biomimetic Hydroxyapatite (HA) was obtained by Chemical Center Srl (Research and Development Department, Italy) and synthesised according to the procedure of Palazzo et al. [34].

### 2.2. Production and Characterisation of HA Loaded with Quercetin Glycoside Compounds and Lactoferrin

Based on previous antimicrobial activity results of the LACTO and QUE complexes against *Pseudomonas fluorescent* [20], HA/LACTO-QUE complexes were prepared with 100 mg/L in LACTO and QUE. LACTO and QUE were adsorbed in HA crystals according to the procedure reported by Montone et al. [25].

The morphology of HA/LACTO-QUE complexes was evaluated by a Scanning Electron 33 Microscope (Leo, model EVO 50). Before the analysis, samples were placed on a conductive graphite surface and then coated with a thin layer of silver in a sputter coater (Edwards, S150B) for 3 min through a flux of argon ions. After this time, the argon flow was stopped, and the samples were left under vacuum for 24 h.

Fourier Transformed Infrared analysis of HA, QUE, LACTO and HA/LACTO-QUE complexes was performed according to Nocerino et al. [37], using a Thermo Nicolet 380 FT-IR spectrometer.

### 2.3. Preparation of Alginate-Based Solution

Alginate solution was prepared by dissolving sodium alginate in distilled water according to the procedure described by Malvano et al. [7]. Successively, two types of sodium alginate solutions were prepared: the first one contained 100 ppm of LACTO and QUE, and the second one HA/LACTO/QUE complexes with 100 ppm of LACTO and QUE.

#### Quercetin and Lactoferrin Release from Coatings

The release study of quercetin and lactoferrin from alginate-based coatings charged with HA/LACTO, HA/QUE and HA/LACTO-QUE complexes was performed following the procedure developed in our previous studies [24,25].

### 2.4. Preparation of Coatings and Study of Pork Meat Fillets Storage

#### 2.4.1. Pork Meat Fillets Coatings and Storage Study

Fresh pork meat fillets (60–80 g) were obtained from whole pork meat piece using an automatic slicer. The edible coating process was carried out by exploiting the layer-by-layer method. Pork fillet samples were dipped before into sodium alginate solutions for 1 min and then in calcium chloride solution (1.5% *w*/*v*) for 1 min, according to Malvano et al. [7]. After, the samples were got dried at room temperature for 15 min and then put into PET boxes provided with a lid. Storage study was performed by comparing three different pork meat samples prepared as below:Uncoated Pork meat fillets (C)Pork meat fillets covered by alginate solution charged with free LACTO and QUE (LACTO-QUE)Pork meat fillets covered by alginate solution charged with HA/LACTO-QUE complexes (HA/LACTO-QUE)

Fifteen boxes for each treatment were maintained at 4 °C for 15 days. At 0, 2, 4, 7, 11 and 15 days physicochemical, microbiological, texture and sensory analyses were carried out. For each storage time, three replicate pork fillet boxes were employed.

#### 2.4.2. Microbiological Analysis

The microbiological parameters investigated during the storage of uncoated and coated pork fillets samples were Total viable bacterial count (TVC), Psychrotrophic bacteria count (PBC), *Pseudomonas* spp. and *Enterobacteriaceae.* The analysis of the above microbiological parameters was performed according to Malvano et al. [7].

#### 2.4.3. Total Volatile Basic Nitrogen Evaluation

Total Volatile Basic Nitrogen (TVB-N) was evaluated by a Kjeldahl distillation unit (UDK139 Velp Scientifica) according to Albanese et al. [38].

#### 2.4.4. Water-Holding Capacity Evaluation

Water-holding capacity (WHC) was calculated as a percentage of weight loss concerning its initial weight, according to the Equation (1):(1)$$\mathrm{WHC}=\frac{\mathrm{Initial}\;\mathrm{Weight}\;\left({\mathrm{day}}_{0}\right)-\mathrm{Weight}\;\left(\mathrm{analysis}\;\mathrm{day}\right)}{\mathrm{Initial}\;\mathrm{Weight}\;\left({\mathrm{day}}_{0}\right)}*100$$

Coatings were manually removed from pork samples and the weight of the samples was recorded.

#### 2.4.5. pH and Colour Evaluation

The pH values, as well as total colour differences (ΔE), were determined according to Malvano et al. [7].

#### 2.4.6. Texture Profile Analysis (TPA)

A Texture Analyzer (LRX Plus, Lloyd Instruments, Chicago) provided with a 100 N load cell was employed to evaluate Hardness (N), Gumminess (N), Cohesiveness (dimensionless), Chewiness (N*mm) and Springiness (mm) parameters. Two consecutive compressions with a cylindrical probe of 1 cm diameter at 1 mm/min were performed on the pork fillet samples. All measurements were performed in triplicate for each fillet sample.

#### 2.4.7. Sensory Evaluation

Coated (LACTO-QUE; HA/LACTO-QUE) and control (C) pork samples were assessed by colour and odour before the cooking, and after the cooking, performed by broiling, for taste, odour and overall acceptability. The active alginate-based coatings were not removed from the samples to evaluate their impact on the colour, odour and taste before and after the cooking of pork fillets.

Ten members (4 male and 6 female) of the Department of Industrial Engineering at the University of Salerno (Italy) were engaged, based on their frequency of consumption of pork meat. Before the sensory trials, all Panel Participants released signed Informed Consent according to American Meat Science Association [39].

The sensory attributes were rated on a 5-point scale from “none” (1) to “high” (5). Scores equal to or lower than 3 were considered not suitable for marketing.

#### 2.4.8. Statistical Analysis

All the analyses were performed in triplicates. Experimental data were reported as mean and standard deviation and subjected to analysis of variance (ANOVA). The significance of differences (*p* < 0.05) among samples was determined by Student’s 𝑡-test with SPSS software version 13.0 for Windows (SPSS, Inc., Chicago, IL, USA).

## 3. Results and Discussion

### 3.1. HA Complex Characterisation

FT-IR spectra of HA, LACTO, QUE and HA/LACTO-QUE complexes are reported in Figure 1.

According to previous studies [7,40] the FT-IR spectra of HA show the characteristic adsorption bands at 1000–1100 cm^−1^ related to the asymmetric stretching mode of vibration for PO_4_ group and other bands at, 880 cm^−1^, 1466 cm^−1^, 1545 cm^−1^ and 3497 cm^−1^ due to the carbonate type A (hydroxyl site)-substituted and type B (phosphate site)-substitute HA crystals.

Moreover, the FT-IR analysis of HA/LACTO-QUE complexes revealed both active compounds. In particular, QUE showed its characteristics peaks at 1500 cm^−1^ (corresponding to C=C stretching), 1670 cm^−1^ (corresponding to C=O stretching), 3248 cm^−1^ (corresponding to O-H stretching) and other peaks in the range of 650 and 1000 cm^−1^ pointed out the presence of the aromatic compounds [7]. LACTO showed its characteristic peaks at 1655 cm^−1^ (C=O), 1532 cm^−1^ (C-H) and 1392 cm^−1^ (C-N) [41].

SEM images of HA, LACTO, QUE, HA/LACTO and HA/LACTO-QUE complexes were reported in Figure 2.

According to our previous study [7], SEM images of HA showed μm particle size (Figure 2a). HA nanoparticles tend to agglomerate in micrometric clusters probably due to Ostwald ripening [42] and their Z-potential values near 0 mV [7].

The characteristic strip-like structures of quercetin glycoside compounds [43] and the spherical structures of lactoferrin [44] are clearly shown in Figure 2b and Figure 2c respectively. SEM images of HA complexes highlighted the first doping with LACTO (Figure 2d) and after with QUE (Figure 2e) showing the characteristic structures of QUE and LACTO, respectively, in the crystal structure.

### 3.2. Release Study

The release of a bioactive compound from a biopolymer matrix into an aqueous medium happens thanks to physical phenomena occurring in sequence and is ruled by physicochemical interactions between polymer, solvent and the bioactive compound. At first, the water penetrates and diffuses into the structure causing the polymeric network widens which allows the diffusion of the active compound until the equilibrium is reached [45].

The kinetic releases of QUE and LACTO from alginate-based coating loaded with HA/QUE, HA/LACTO and HA/LACTO-QUE complexes are shown in Figure 3. As can be observed for the coating loaded with HA/LACTO-QUE complexes, the release of LACTO, as well as the achievement of equilibrium, occurs in a shorter time than QUE. In particular, the release of LACTO started after 9 h compared with the QUE release that occurred after 31 h. Moreover, QUE reached the equilibrium 42 h later than LACTO.

As regards the influence of simultaneous loading of the active compounds into HA crystals, the presence of QUE seems to influence the release of LACTO, which is faster in the case of HA/LACTO-QUE complexes than in HA/LACTO ones. In contrast, the coating loaded with HA/QUE complexes starts releasing quercetin in a shorter time before than the coating loaded with HA/LACTO-QUE ones, even if no kinetic release differences were observed.

The observed different release behaviour in active alginate-based coatings could be due to the difference in the solubility of the active compounds in the aqueous medium (10 g/L for LACTO and 1.61 g/L for QUE) besides the different interactions of the active compounds could show with HA structure.

### 3.3. Changes in Microbial, Chemical-Physical and Sensory Properties of Pork Meat

#### 3.3.1. Microbiological Changes

Microbial changes in the coated and uncoated fresh pork meat are reported in Figure 4. For all microbial parameters, an increase was measured at the increase of the storage time.

The initial TVC of pork meat samples (Figure 4a) was 3.50 log CFU/g, according to previous studies on the shelf-life evaluation of pork meat [14,17].

TVC values increased with the increase of storage days for coated and uncoated samples. On day 7, uncoated samples reached TVC values close to 7 log CFU/g, considered the maximum acceptable limit for fresh meat [46].

LACTO-QUE coated samples reached the TVC threshold at 11 days while this value was never reached by HA/LACTO-QUE samples until the end of the storage period. This indicates that, from a microbiological point of view, a shelf-life extension, of at least 7 and 4 days was obtained for HA/LACTO-QUE samples compared with the control and HA/LACTO, respectively.

Coating loaded with HA/LACTO-QUE complexes inhibited the growth of the Enterobacteriaceae (Figure 4b) and the final charge (4.6 log CFU/g), at 15 storage days, was lower (1 log cycle) significantly (*p* < 0.05) compared to the control sample.

Meat spoilage is mainly caused by the metabolic activity of psychrotrophic bacteria, in particular *Pseudomonas* spp., the prevailing spoilage flora of food from the animal origin [21] causing off-odours and off-flavours during storage in cold conditions. Figure 3c,d show that PBC and *Pseudomonas* spp. for uncoated and coated pork meat samples showed a similar trend to the TVC, increasing with the increase of storage time.

In particular, during the storage period, LACTO-QUE and HA/LACTO-QUE samples showed significantly (*p* < 0.05) lower *Pseudomonas* spp. and PTC, as compared to the control.

The results of LACTO-QUE and HA/LACTO-QUE coated samples highlighted the higher capability of the coating to slow down bacterial growth when the active compounds were adsorbed into HA crystals. The antimicrobial activity of both quercetin glycoside compounds and lactoferrin against meat spoilage microorganisms such as *Pseudomonas* spp. has been tested by our group and other authors [24,37,47,48,49]. In addition, our group also verified the enhanced antibacterial activity of lactoferrin and quercetin when adsorbed into HA crystals [25]. The capability of alginate-based coating charged with HA/LACTO-QUE to slow down the microbial charge coated pork fillets during the storage agrees with our previous study focused on synergistic inhibition effect versus *Pseudomonas fluorescens* of LACTO and QUE loaded in the structure of hydroxyapatite [25]. When HA was incubated firstly with LACTO and then with QUE, the highest inhibition was reached. The complex HA/LACTO-QUE (100 pm) exhibited a synergistic effect on the growth of *Pseudomonas fluorescens*, showing a fractional inhibitory concentration (FIC) index equal to 0.4, according to the interpretation that a FIC index of ≤0.5 suggests the synergistic interaction [50].

#### 3.3.2. pH, Total Volatile Basic Nitrogen and Colour Evaluation

Meat pH strongly influenced meat quality and freshness characteristics, such as colour, tenderness and microbial growth [51]. The effects of alginate-based coatings on pork meat pH during the storage period are shown in Figure 5a.

The pH value of the fresh pork meat sample was 5.59, close to the values reported by other authors [14,16,17], and it increased with the increase of the storage days in all samples. In particular, until day 4 no pH change was observed in HA/LACTO-QUE samples. On day 11, a significant increase in pH was observed for all meat samples: however, the pH values were always lower in the coated samples than in the control ones. At the end of the storage, the coating containing HA/LACTO-QUE complexes granted the lowest pH value, with an increase of 21.82% concerning day 0. An increase of 30.59% and 27.19% was reached for the control and LACTO-QUE samples, respectively, at the end of the storage period.

The rise of pH in pork samples may be explained as related to the degradation of proteins that occur at the last stage of storage that allows the production of volatile alkaline nitrogen molecules, including ammonia and amines, through microbial activity and meat endogenous proteases [14,38]. pH results agreed with the amount of total volatile basic nitrogen (TVB-N), which represents the main product of protein decomposition by spoilage bacteria in pork meat [52] affecting the sensory acceptability of the meat besides being toxic for human health [53].

Changes in TVB-N values for all fillet pork samples are reported in Figure 4b. The initial value of fresh meat was 1.15 mg/100 g, and this value increased with the storage time. In fact, according to microbiological results, microbial growth causes protein degradation and damage to the muscle cell structure. As result, endogenous enzymes were released that accelerated protein degradation and the release of amino acids [54].

On each day of storage, the TVB-N values of uncoated samples were significantly (*p* < 0.05) higher than coated ones. The minimumTVB-N value was guaranteed, at any time of storage, by the coating containing hydroxyapatite–lactoferrin–quercetin complexes. Considering that the TVB-N acceptable threshold for pork meat is ≤15 mg/100 g [55], control and LACTO-QUE samples exceeded the limit on 11th cold storage, in contrast to the HA/LACTO-QUE samples, which did not reach this limit for the whole storage period.

Changes in L*, a*, b* and total colour differences ΔE of pork meat samples during the storage time are shown in Table 2. At the beginning of storage (day 0), coated samples showed similar lightness, lower than the control sample. This difference could be due to the colour of the film-forming solution, which is the result of the pale-yellow colour of sodium alginate adding to the yellow and pink colour of quercetin and lactoferrin, respectively.

L* values of coated and uncoated pork meat samples remained constant until day 7, and after that a rapid decrease in lightness was registered for uncoated samples. The pork meat coated samples showed a slight decrease in lightness from day 7 until the end of storage. This means that the alginate-based coating was able to protect meat oxygen responsible for browning phenomena, while active compounds as antioxidants reduced oxidation. Similar results were obtained also by Ruan et al. [14]. The initial a* value, which indicates the freshness of pork meat, ranged from 4.40 to 7.20 and these values did not change until day 2. After that, the coated samples showed a slight increase during the entire storage period, while a marked increase was registered for uncoated ones.

b* values increased during the storage period (Table 2), probably due to the presence of hydrogen sulfide (H_2_S) produced by microorganisms and enzymes that degraded proteins and bind to haemoglobin to form yellow complexes [55].

In contrast to uncoated samples that increase quickly from day 2 to the end of the storage period, a slight b* increase was observed for coated samples, thanks to the beneficial effect of coating against microbial spoilage. Moreover, the addition of hydroxyapatite in alginate-based edible coating had a better colour protection effect on fresh pork meat. Finally, a constant increase in total colour was observed during the cold storage period for coated pork samples, while a significantly (*p* < 0.05) higher ΔE value was obtained for uncoated ones.

#### 3.3.3. WHC and TPA

Water-holding capacity (WHC) is an important attribute related to fresh meat quality, influencing the freshness, cooking yield and sensory palatability, as well as nutritional profile [56].

As reported in Table 2, water losses gradually increased for all pork samples during storage time, ranging from 1.41% to 12.03% for uncoated samples and from 0.93% to 9.43% and from 0.41% to 8.06% for LACTO-QUE and HA/LACTO-QUE samples, respectively. This behaviour could be due to the water-barrier properties of alginate-based coatings that prevent exudation and meat dehydration [5]. The comparison between coated pork samples pointed out a significantly (*p* < 0.05) higher WHC in HA/LACTO-QUE than coatings loaded with free quercetin and lactoferrin. These results agree with our previous studies [7,35], highlighting the capability of HA structure to lead to a strong water-holding capacity.

Texture parameters measured of coated and uncoated pork samples are reported in Table 3. As can be observed, the alginate layer on the surface of pork fillets influences the texture parameters of pork fillets showing significant differences in comparison with uncoated samples. The hardness increased significantly at 15 storage days only in control and LACTO-QUE samples. As previously stated, the increased hardness in meat products may be correlated to water leakage because of the evaporation of water from the meat surface and the reduced capability of meat proteins to hold water [13]. The decrease in springiness values was registered at the end of the storage period for all pork fillet samples. The lowering in springiness could be explained by a lower-elasticity meat structure arising from water leaks. Values of cohesiveness, gumminess and chewiness did not show significant (*p* < 0.05) differences among samples during the cold storage.

#### 3.3.4. Sensory Evaluation

Changes in sensory properties (colour, odour, taste and overall acceptability) of coated (LACTO-QUE; HA/LACTO-QUE) and control (C) pork meat samples before and after cooking are shown in Table 4. The scores of samples showed a similar downward trend that, as expected, decreased with the increase in storage time. On the fourth day, the colour of uncoated samples was significantly (*p* < 0.05) lower than coated pork samples, reaching unacceptable scores on the seventh day. Between coated pork samples, LACTO-QUE reached a score of 3 on the 15th day, while HA/LACTO-QUE samples remained acceptable during the whole storage period.

Until day 4 there were no significant (*p* < 0.05) differences between coated and control pork fillet samples, for both raw and cooked pork fillets. However, from the seventh day, the odour score of coated samples was significantly (*p* < 0.05) higher than those of uncoated ones. At the end of the storage period, HA/LACTO-QUE samples showed values higher than the acceptability threshold. The overall acceptability of uncoated cooked pork samples was insufficient on the seventh day of storage.

The coating with alginate coating charged with LACTO and QUE allowed pork fillets to reach overall acceptability scores of up to 11 storage days, while the addition of HA/LACTO-QUE made pork samples acceptable for the whole investigated storage period. These results were consistent with the values of TVB-N and TVC reported in the sections discussed above. As regards the taste evaluated at time 0 for the cooked pork fillets no significant differences (*p* < 0.05) were observed among the control and coated samples. In particular, the cooking, due to the high temperature, caused the thermal breakdown of the alginate coating, though characterised by a neutral taste [7].

## 4. Conclusions

An alginate-based coating activated with hydroxyapatite/lactoferrin/quercetin complexes was developed and applied to fresh pork meat to extend its shelf life.

The morphological analysis confirmed the adsorption of both bioactive compounds into the hydroxyapatite network, while in-vitro studies showed a homogeneous release of quercetin glycoside compounds and lactoferrin through the coating, reaching equilibrium in 70 h and 30 h, respectively.

The activated alginate-based coating showed a high capability to slow down the growth of the main microorganism responsible for the spoilage of fresh meat products, during storage for 15 days at 4 °C, reducing the production of the volatile basic nitrogen compounds.

Moreover, the comparison among samples pointed out a positive effect of the coating charged with hydroxyapatite/lactoferrin/quercetin complexes to slow down the changes in hardness during the storage time as well as the sensory attributes for both uncooked and cooked pork fillets. Finally, the results of the sensory evaluation showed that the presence of edible coating did not affect the visual and taste attributes in raw and cooked fillets, making the proposed active edible coating a potential application for the shelf-life extension of fresh meat products.

## Figures and Tables

**Figure 1 foods-12-00553-f001:**
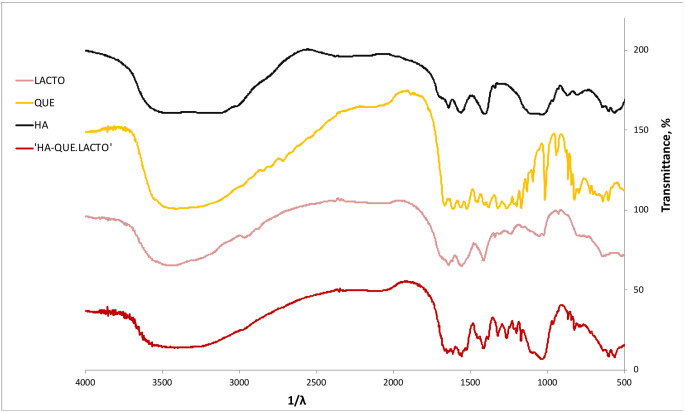
FT-IR spectra of HA, LACTO, QUE and HA/LACTO-QUE complexes.

**Figure 2 foods-12-00553-f002:**
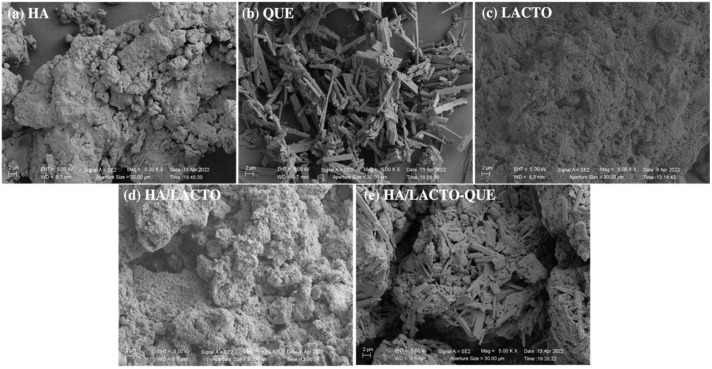
SEM images of HA (**a**), QUE (**b**), LACTO (**c**) HA/LACTO complexes (**d**) and HA/LACTO-QUE (**e**) complexes.

**Figure 3 foods-12-00553-f003:**
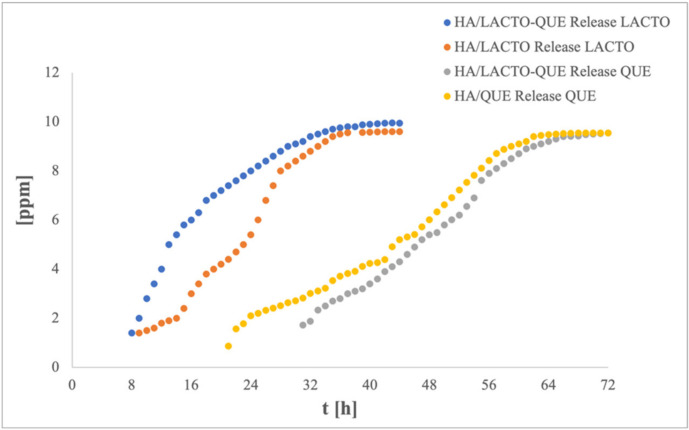
Release profile of QUE and LACTO from HA complexes.

**Figure 4 foods-12-00553-f004:**
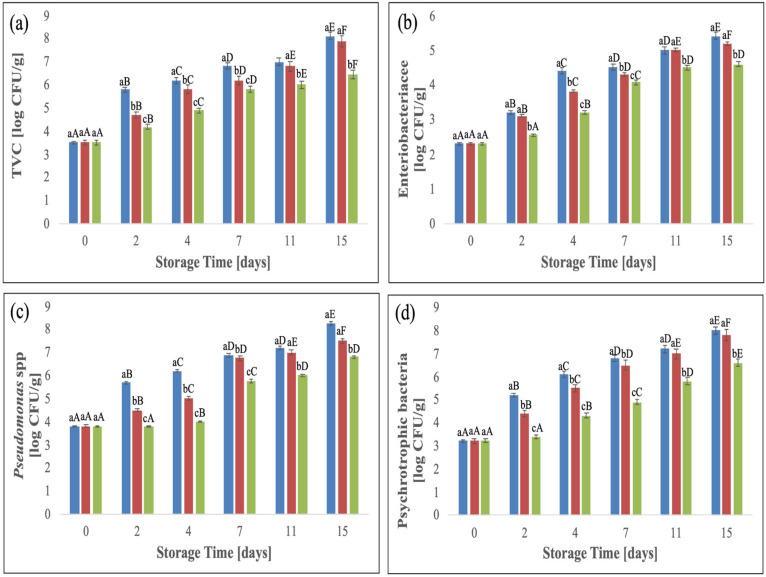
Changes in total viable bacterial count (**a**), *Enterobacteriaceae* (**b**), *Pseudomonas* spp. (**c**) and psychrotrophic bacteria count (**d**) of pork meat samples during the storage time at 4 °C. Different letters (a, b, c,…) reveal significant differences (*p* < 0.05) among the samples for each storage time, and different letters (A, B, C,…) reveals significant differences (*p* < 0.05) for each treatment during the storage time.

**Figure 5 foods-12-00553-f005:**
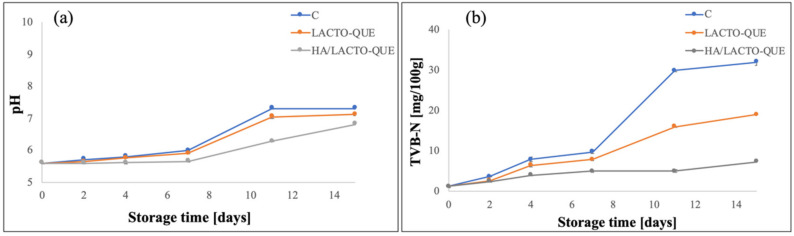
Effects of alginate-based coatings on pH (**a**) and TVB-N (**b**) of pork meat samples during the storage time.

**Table 1 foods-12-00553-t001:** Recent applications of active alginate-based coatings in fresh meat and different types of meat products.

Food	Active Substance	Results	Reference
Chicken meat	Quercetin glycoside compounds	Coating significantly inhibited the growth of spoilage bacteria, as well as the total volatile basic nitrogen and slowed down the changes in hardness during the storage time of 11 days.	[7]
Chicken meat	Citrus and Lemon	Coating resulted in less growth of microorganisms in the samples.	[18]
Chicken meat	Black cumin	High antimicrobial activity versus *Escherichia coli*, less variation in pH and lower colour changes, over 5 days of storage at 4 °C.	[19]
Chicken meat	Lactoperoxidase	Lactoperoxidase addition into the alginate-based coating system led to higher bacterial and sensorial quality values of chicken meat. The effect was even increased with the increasing concentration of lactoperoxidase.	[20]
Beef	Ginger essential oils	Active coating increased the shelf life of chilled beef slices by 9 days, delaying lipid oxidation and microbial spoilage.	[21]
Beef	Basil	The active coating increased antioxidant activity and reduced meat lipid oxidation and weight loss.	[22]
Pork Meat	epigallocatechin gallate	The results showed that fresh pork coated with the active coating had a significant inhibitory effect on its microbial growth.	[14]
Lamb meat	Essential oils of thyme and garlic	The active coating had an effect on lamb meat quality that helps maintain its characteristics during its shelf life after thawing. Thyme led to lower lipid oxidation and better colour maintenance.	[13]

**Table 2 foods-12-00553-t002:** Effects of alginate-based coatings on the colour of pork meat samples during the cold storage.

Parameter	Sample	Storage Time (Days)					
		0	2	4	7	11	15
L*	C	68.84 ± 0.69 ^aA^	68.84 ± 0.65 ^aA^	67.95 ± 0.68 ^aA^	65.95 ± 0.66 ^aB^	52.33 ± 0.52 ^aC^	50.16 ± 0.50 ^aD^
	QUE-LACTO	63.88 ± 0.64 ^bA^	62.88 ± 0.63 ^bA^	62.82 ± 0.62 ^bA^	60.89 ± 0.61 ^bB^	59.78 ± 0.60 ^bB^	58.04 ± 0.58 ^bC^
	HA/QUE-LACTO	64.84 ± 0.65 ^bA^	64.84 ± 0.65 ^cA^	63.95 ± 0.64 ^bA^	62.45 ± 0.62 ^cB^	61.23 ± 0.61 ^cC^	59.92 ± 0.60 ^cD^
a*	C	4.40 ± 0.04 ^aA^	4.40 ± 0.03 ^aA^	5.68 ± 0.06 ^aB^	7.27 ± 0.07 ^aC^	11.76 ± 0.12 ^aD^	13.50 ± 0.13 ^aE^
	QUE-LACTO	7.12 ± 0.07 ^bA^	7.12 ± 0.07 ^bA^	7.30 ± 0.07 ^bB^	8.78 ± 0.09 ^bC^	9.87 ± 0.09 ^bD^	12.15 ± 0.12 ^bE^
	HA/QUE-LACTO	7.20 ± 0.07 ^cA^	7.20 ± 0.06 ^bA^	7.60 ± 0.08 ^cB^	8.78 ± 0.09 ^bC^	10.20 ± 0.10 ^cD^	11.56 ± 0.12 ^cE^
b*	C	3.25 ± 0.03 ^aA^	3.25 ± 0.02 ^aA^	4.08 ± 0.04 ^aB^	5.54 ± 0.05 ^aC^	7.01 ± 0.07 ^aD^	7.26 ± 0.07 ^aE^
	QUE-LACTO	3.79 ± 0.04 ^bA^	3.79 ± 0.03 ^bA^	3.84 ± 0.04 ^bA^	4.20 ± 0.04 ^bB^	4.59 ± 0.05 ^bC^	5.36 ± 0.05 ^bD^
	HA/QUE-LACTO	3.09 ± 0.03 ^cA^	3.09 ± 0.03 ^cA^	3.20 ± 0.03 ^cB^	3.60 ± 0.02 ^cC^	4.09 ± 0.04 ^cD^	4.78 ± 0.05 ^cE^
ΔE	C	-	0.00 ± 0.00 ^aA^	1.77 ± 0.01 ^aB^	4.80 ± 0.02 ^aC^	18.46 ± 1.23 ^aD^	21.16 ± 1.78 ^aE^
	QUE-LACTO	-	0.00 ± 0.00 ^aA^	1.57 ± 0.02 ^bB^	2.62 ± 0.03 ^bC^	4.22 ± 0.51 ^bD^	7.10 ± 0.34 ^bE^
	HA/QUE-LACTO	-	0.00 ± 0.00 ^aA^	0.98 ± 0.01 ^cB^	2.91 ± 0.02 ^cC^	4.80 ± 0.23 ^cD^	6.78 ± 0.48 ^cE^

Different letters (a, b, c,…) reveal significant differences (*p* < 0.05) among the samples for each storage time, and different letters (A, B, C,…) reveal significant differences (*p* < 0.05) for each treatment during the storage time.

**Table 3 foods-12-00553-t003:** Effects of alginate-based coatings on WHC and textural parameters of pork meat samples during the cold storage.

		Storage Time (Days)					
Parameter	Sample	0	2	4	7	11	15
WHC	C	-	1.41 ± 0.02 ^aA^	4.73 ± 0.03 ^aB^	6.69 ± 0.08 ^aC^	11.12 ± 1.12 ^aD^	12.04 ± 0.98 ^aE^
	LACTO-QUE	-	0.93 ± 0.01 ^bA^	3.61 ± 0.02 ^bB^	5.83 ± 0.04 ^bC^	7.44 ± 0.05 ^bD^	9.43 ± 0.78 ^bE^
	HA/LACTO-QUE	-	0.41 ± 0.01 ^cA^	1.36 ± 0.02 ^cB^	3.59 ± 0.03 ^cC^	5.97 ± 0.06 ^cD^	8.06 ± 0.56 ^cE^
Hardness [N]	C	10.25 ± 0.30 ^aA^	13.02 ± 0.50 ^aB^	12.77 ± 0.50 ^aB^	12.31 ± 0.70 ^aB^	13.35 ± 0.10 ^aC^	15.66 ± 0.10 ^aD^
	LACTO-QUE	13.03 ± 0.40 ^bA^	13.83 ± 0.60 ^aA^	12.54 ± 0.60 ^aB^	12.70 ± 0.50 ^aB^	13.28 ± 0.60 ^aB^	14.83 ± 0.10 ^bC^
	HA/LACTO-QUE	13.27 ± 0.28 ^bA^	13.27 ± 0.30 ^aA^	12.45 ± 0.60 ^aA^	12.98 ± 0.20 ^aA^	12.62 ± 0.60 ^aA^	13.09 ± 0.50 ^cA^
Cohesiveness	C	0.03 ± 0.01 ^aA^	0.03 ± 0.01 ^aA^	0.01 ± 0.01 ^aA^	0.02 ± 0.00 ^aA^	0.02 ± 0.01 ^aA^	0.02 ± 0.01 ^aA^
	LACTO-QUE	0.03 ± 0.00 ^aA^	0.03 ± 0.00 ^aA^	0.03 ± 0.01 ^aA^	0.02 ± 0.01 ^aA^	0.03 ± 0.00 ^aA^	0.02 ± 0.01 ^aA^
	HA/LACTO-QUE	0.03 ± 0.00 ^aA^	0.03 ± 0.00 ^aA^	0.02 ± 0.01 ^aA^	0.02 ± 0.01 ^aA^	0.02 ± 0.01 ^aA^	0.03 ± 0.00 ^aA^
Springiness [mm]	C	0.35 ± 0.01 ^aA^	0.40 ± 0.06 ^aA^	0.43 ± 0.08 ^aA^	0.44 ± 0.06 ^aA^	0.42 ± 0.05 ^aA^	0.25 ± 0.01 ^aB^
	LACTO-QUE	0.38 ± 0.01 ^bA^	0.39 ± 0.00 ^aA^	0.42 ± 0.07 ^aA^	0.44 ± 0.06 ^aA^	0.36 ± 0.07 ^aA^	0.34 ± 0.01 ^bB^
	HA/LACTO-QUE	0.38 ± 0.00 ^bA^	0.36 ± 0.03 ^aA^	0.35 ± 0.07 ^aA^	0.39 ± 0.07 ^aA^	0.38 ± 0.04 ^aA^	0.36 ± 0.02 ^bA^
Gumminess [N]	C	0.37 ± 0.00 ^aA^	0.37 ± 0.10 ^aA^	0.22 ± 0.10 ^aA^	0.30 ± 0.10 ^aA^	0.40 ± 0.10 ^aA^	0.36 ± 0.10 ^aA^
	LACTO-QUE	0.38 ± 0.00 ^bA^	0.38 ± 0.10 ^aA^	0.37 ± 0.10 ^A^	0.25 ± 0.10 ^aA^	0.33 ± 0.05 ^aA^	0.27 ± 0.00 ^aA^
	HA/LACTO-QUE	0.39 ± 0.00 ^bA^	0.39 ± 0.10 ^aA^	0.37 ± 0.10 ^A^	0.23 ± 0.10 ^aA^	0.20 ± 0.10 ^aA^	0.33 ± 0.10 ^aA^
Chewiness [N*mm]	C	0.11 ± 0.00 ^aA^	0.13 ± 0.00 ^aB^	0.09 ± 0.00 ^aC^	0.13 ± 0.00 ^aD^	0.12 ± 0.00 ^aE^	0.08 ± 0.00 ^aF^
	LACTO-QUE	0.14 ± 0.00 ^bA^	0.13 ± 0.00 ^aB^	0.15 ± 0.10 ^aB^	0.06 ± 0.02 ^bB^	0.12 ± 0.10 ^aB^	0.09 ± 0.00 ^bC^
	HA/LACTO-QUE	0.14 ± 0.00 ^bA^	0.14 ± 0.00 ^bA^	0.09 ± 0.00 ^aB^	0.08 ± 0.00 ^bC^	0.07 ± 0.10 ^aC^	0.12 ± 0.00 ^cD^

Different letters (a, b, c,…) reveal significant differences (*p* < 0.05) among the samples for each storage time, and different letters (A, B, C,…) reveal significant differences (*p* < 0.05) for each treatment during the storage time.

**Table 4 foods-12-00553-t004:** Effects of alginate-based coatings on sensory parameters of pork meat samples during the cold storage.

	Parameter	Sample	Storage Time (Days)					
			0	2	4	7	11	15
Fresh pork meat	Colour	C	5.00 ± 0.00 ^aA^	4.80 ± 0.20 ^aA^	4.00 ± 0.10 ^aB^	2.40 ± 0.40 ^aC^	1.10 ± 0.10 ^aD^	1.00 ± 0.00 ^aD^
		LACTO-QUE	5.00 ± 0.00 ^aA^	4.60 ± 0.40 ^aA^	4.40 ± 0.20 ^bA^	3.80 ± 0.20 ^bB^	3.20 ± 0.18 ^bC^	3.00 ± 0.20 ^bC^
		HA/LACTO-QUE	5.00 ± 0.00 ^aA^	4.80 ± 0.20 ^aA^	4.60 ± 0.20 ^bA^	4.00 ± 0.60 ^bA^	3.60 ± 0.20 ^cB^	3.40 ± 0.10 ^cB^
	Odour	C	5.00 ± 0.00 ^aA^	4.80 ± 0.40 ^aA^	4.40 ± 0.40 ^aA^	3.40 ± 0.10 ^aB^	1.40 ± 0.20 ^aC^	1.00 ± 0.00 ^aD^
		LACTO-QUE	5.00 ± 0.00 ^aA^	5.00 ± 0.00 ^aA^	4.60 ± 0.20 ^aB^	4.40 ± 0.10 ^bB^	3.80 ± 0.10 ^bC^	3.00 ± 0.10 ^bD^
		HA/LACTO-QUE	5.00 ± 0.00 ^aA^	5.00 ± 0.00 ^aA^	4.60 ± 0.20 ^aB^	4.40 ± 0.20 ^bB^	3.80 ± 0.20 ^bC^	3.60 ± 0.20 ^cC^
Cooked pork meat	Taste	C	5.00 ± 0.00 ^aA^	4.60 ± 0.20 ^aB^	3.40 ± 0.10 ^aC^	2.00 ± 0.00 ^aD^	1.40 ± 0.20 ^aE^	1.00 ± 0.00 ^aF^
		LACTO-QUE	5.00 ± 0.00 ^aA^	4.80 ± 0.40 ^aB^	4.40 ± 0.40 ^bB^	3.80 ± 0.10 ^bC^	3.60 ± 0.10 ^bC^	3.00 ± 0.20 ^bD^
		HA/LACTO-QUE	5.00 ± 0.00 ^aA^	5.00 ± 0.00 ^aA^	4.60 ± 0.20 ^bB^	4.40 ± 0.20 ^cB^	4.00 ± 0.00 ^cC^	3.80 ± 0.20 ^cC^
	Odour	C	5.00 ± 0.00 ^aA^	4.80 ± 0.20 ^aA^	4.60 ± 0.20 ^aA^	3.40 ± 0.10 ^aB^	1.80 ± 0.40 ^aC^	1.20 ± 0.10 ^aD^
		LACTO-QUE	5.00 ± 0.00 ^aA^	5.00 ± 0.00 ^aA^	4.60 ± 0.20 ^aB^	4.40 ± 0.10 ^bB^	3.00 ± 0.20 ^bC^	1.80 ± 0.20 ^bD^
		HA/LACTO-QUE	5.00 ± 0.00 ^aA^	5.00 ± 0.00 ^aA^	4.60 ± 0.20 ^aB^	4.60 ± 0.20 ^bB^	3.80 ± 0.20 ^cC^	3.60 ± 0.20 ^cC^
	Overall acceptability	C	5.00 ± 0.00 ^aA^	4.80 ± 0.20 ^aA^	4.20 ± 0.30 ^aB^	3.00 ± 0.20 ^aC^	1.20 ± 0.20 ^aD^	1.00 ± 0.10 ^aD^
		LACTO-QUE	5.00 ± 0.00 ^aA^	4.80 ± 0.20 ^aA^	4.60 ± 0.30 ^aA^	4.20 ± 0.10 ^bB^	3.80 ± 0.10 ^bC^	3.00 ± 0.10 ^bD^
		HA/LACTO-QUE	5.00 ± 0.00 ^aA^	4.80 ± 0.20 ^aA^	4.60 ± 0.60 ^aA^	4.40 ± 0.10 ^bA^	3.80 ± 0.10 ^bB^	3.80 ± 0.10 ^cB^

Different letters (a, b, c,…) reveal significant differences (*p* < 0.05) among the samples for each storage time, and different letters (A, B, C,…) reveal significant differences (*p* < 0.05) among treatments during the storage time.

## Data Availability

The data presented in this study are available on request from the corresponding author.

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
