# Peer review of "Alginate Coating Charged by Hydroxyapatite Complexes with Lactoferrin and Quercetin Enhances the Pork Meat Shelf Life"

_foods, 2023, doi:10.3390/foods12030553_

Round 1

Author Response

Answers to Reviewer 1

Manuscript foods-2115063: Alginate coating charged by Hydroxyapatite complexes with lactoferrin and quercetin enhances the pork meat shelf life

We gratefully acknowledge the Reviewer for his thorough and careful examination of the manuscript. We have considered all his comments/suggestions and revised the paper accordingly. Additions in the manuscript are marked in red

The manuscript contains a great deal of work, which studies the effect of the coating with alginate as the base material and loaded with hydroxyapatite compound adsorbed with lactoferrin and quercetin on the shelf life of fresh pork. Hydroxyapatite is rarely used as food packaging. According to the research of manuscripts, it can be used as a good carrier of active compounds. There are some details that are missing and questions that need to be answered. I suggest that the author minor revise the paper. The specific opinions are as follows

  1. Line41chiken

Done, see the revised manuscript

  1. Line131Add ", "after "In particular".

We have added the comma after “In particular, see the revised manuscript

  1. Line163How to Remove Coating from Pork Samples?

Coatings were manually removed from pork samples before analysis of Water Holding Capacity. In the revised manuscript (line 97) we have described how.

  1. Line183Has the coating on the raw pork sample been removed?

The active alginate-based coatings were not removed from the samples to evaluate their impact on the colour, odour and taste before and after the cooking of pork fillets. Based on the above, in the revised manuscript, the section “2.4.7 Sensory evaluation” has been modified as follows: “Coated (LACTO-QUE; HA/LACTO-QUE) and control (C) pork samples were assessed by colour and odour before the cooking, and after the cooking, performed by broiling, for taste, odour and overall acceptability. The active alginate-based coatings were not removed from the samples to evaluate their impact on the colour, odour and taste before and after the cooking of pork fillets.  “. Lines 113-117 of the revised manuscript.

Will it affect the sensory evaluation if it is not removed?

In the results and discussions of the revised manuscript (lines 366-369) we have pointed out the differences regarding the evaluated sensory attributes among coated and control pork fillet samples, for both raw and cooked pork fillets. Shortly, significant differences among control and fillets coated with alginate-based active coatings were observed starting from the fourth storage day. 

  1. Line183Is the pork sample processed before cooking? For example, if the coating is removed, if the coating is not removed, will the chemical reaction of the coating occur again in the cooking process, thus affecting the sensory evaluation?

See the previous response. 

  1. Line199In my opinion, the infrared spectrum of LACTO - QUE is lacking?

The production of HA/LACTO-QUE complexes requires the adsorption in the hydroxyapatite structure of lactoferrin and, subsequently, of quercetin. For this reason, there are no LACTO-QUE IR spectra without hydroxyapatite. 

  1. Line438Will the intake of coating have an impact on human body?

Additional information on human safety related to coating materials (HA, quercetin and lactoferrin) has been added in the revised manuscript (lines 82-88; 105-106) 

  1. Line449were

Done, see the revised manuscript 

As suggested, the English has been improved and edited by an English speaker.

Reviewer 2 Report

The present study aims at evaluating the shelf-life of pork meat samples coated with alginate coating charged by hydroxyapatite complexes with 3 lactoferrin and quercetin. The authors conducted a comprehensive study in which they made microbiological, physio-chemical and sensorial analysis.

 Fresh pork meat originating from Italy was coated with the coatings developed and analyzed over a period of 15 days. Besides the microbiological and physio-chemical analysis on the pork meat, the study also made complexes characterization on the hydroxyapatite (FT-IR and SEM)..

I believe that the paper is well written overall and the methods are carefully described. The objectives and the design of the study are clearly stated and identified in the Introduction section. The authors provided the interpretation of the obtained results in a well-structured manner and as such, I believe the topic of the present paper is relevant and of interest for the readers of the journal.

On the other hand, the introduction could be improved by adding more literature reviews on the effects of alginate coatings on different food products (different types of meat products), or maybe add a table in which the alginate coatings effects on the shelf-life are displayed.

In the “2.1. Materials” please state at what temperatures were the pork meat samples transported from the butchery to the laboratory.

The conclusion part could also be improved given the multiple analysis carried out in the study.

Therefore, I recommend undergoing minor revision.

Author Response

Answers to Reviewer 2 
Manuscript foods-2115063: Alginate coating charged by Hydroxyapatite complexes with lactoferrin and quercetin enhances the pork meat shelf life 

We gratefully acknowledge the Reviewer for his thorough and careful examination of the manuscript. We have considered all his comments/suggestions and revised the paper accordingly. Additions in the manuscript are marked in red

The present study aims at evaluating the shelf-life of pork meat samples coated with alginate coating charged by hydroxyapatite complexes with 3 lactoferrin and quercetin. The authors conducted a comprehensive study in which they made microbiological, physio-chemical and sensorial analysis. 

 Fresh pork meat originating from Italy was coated with the coatings developed and analyzed over a period of 15 days. Besides the microbiological and physio-chemical analysis on the pork meat, the study also made complexes characterization on the hydroxyapatite (FT-IR and SEM).. 

I believe that the paper is well written overall and the methods are carefully described. The objectives and the design of the study are clearly stated and identified in the Introduction section. The authors provided the interpretation of the obtained results in a well-structured manner and as such, I believe the topic of the present paper is relevant and of interest for the readers of the journal. 

On the other hand, the introduction could be improved by adding more literature reviews on the effects of alginate coatings on different food products (different types of meat products), or maybe add a table in which the alginate coatings effects on the shelf-life are displayed.

In the “2.1. Materials” please state at what temperatures were the pork meat samples transported from the butchery to the laboratory.

The conclusion part could also be improved given the multiple analysis carried out in the study.

Therefore, I recommend undergoing minor revision. 

As suggested, in the introduction section of the revised manuscript (line 77) more literature on the positive effects of alginate coatings on meat products were added. In particular, as you suggested, we have added a table in which we have summarized some of the published reports in the last 5 years about the application of active alginate-based coatings in fresh meat and different types of meat products.

Additional information on sample transport temperature was added to the revised manuscript (line 127) and the conclusion section has been improved as suggested.

Finally, the English have been improved and edited by an English speaker.
